
# Vertically resolved concentration and liquid water content of atmospheric nanoparticles at the US DOE Southern Great Plains site

Haihan Chen[1], Anna L. Hodshire[2], John Ortega[3], James Greenberg[3], Peter H. McMurry[4], Annmarie G. Carlton[1], Jeffrey R. Pierce[2], Dave R. Hanson[5], James N. Smith*,[1]

[1]Department of Chemistry, University of California, Irvine, 92697-2025, USA

[2]Department of Atmospheric Science, Colorado State University, Fort Collins, 80523, USA

[3]Atmospheric Chemistry Observations & Modeling Laboratory, National Center for Atmospheric Research, Boulder, 80307, USA

[4]Department of Mechanical Engineering, University of Minnesota-Twin Cities, Minneapolis, 55455, USA

[5]Department of Chemistry, Augsburg College, Minneapolis, 55454, USA

*Correspondence to:* James N. Smith (jimsmith@uci.edu)

**Abstract.** Most prior field studies of new particle formation (NPF) have been performed at or near ground level, leaving many unanswered questions regarding the vertical extent of NPF. To address this, we measured concentrations of 11-16 nm diameter particles from ground level to 1000 m observed during the 2013 New Particle Formation Study at the Atmospheric Radiation Measurement Southern Great Plains site in Lamont,

Oklahoma. The measurements were performed using a tethered balloon carrying two condensation particle counters that were configured for two different particle cut-off diameters. Those observations were compared to data from three scanning mobility particle sizers at the ground level. We observed that 11-16 nm diameter particles were generated at the top region of the boundary layer, and were then rapidly mixed throughout the boundary layer. We also estimate liquid water content of nanoparticles using ground-based measurements of

particle hygroscopicity obtained with a Humidified Tandem Differential Mobility Analyzer and vertically resolved relative humidity (RH) and temperature measured with a Raman Lidar. Our analyses of these observations lead to the following conclusions regarding nanoparticles formed during NPF events at this site: (1) ground-based observations may not always accurately represent the timing, distribution, and meteorological conditions associated with the onset of NPF; (2) nanoparticles are highly hygroscopic, and

typically contain up to 50% water by volume, and during conditions of high RH combined with high particle hygroscopicity, particles can be up to 95% water by volume; (3) increased liquid water content of nanoparticles at high RH greatly enhances the partitioning of water soluble species like organic acids into ambient nanoparticles.



## 1. Introduction

New particle formation (NPF) and growth are frequently observed in the atmosphere worldwide (Finlayson-Pitts and Pitts Jr, 2000; Kulmala et al., 2004; Seinfeld and Pandis, 2006; Zhang et al., 2011). One of the most important potential impacts of NPF is the creation of cloud condensation nuclei (CCN), which

can modify cloud properties and thereby affect climate and precipitation (Kerminen et al., 2005; Pierce and Adams, 2007; Spracklen et al., 2008; Kuang et al., 2009; Merikanto et al., 2009; Pierce and Adams, 2009; Pierce et al., 2012; Westervelt et al., 2013; Farmer et al., 2015). Sulfuric acid is often associated with the formation and growth of new particles (Weber et al., 1996, 1997; Weber et al., 2001; Stolzenburg et al., 2005; Kuang et al., 2008; Sipilä et al., 2010; Kulmala et al., 2013), but other species such as methanesulfonic acid

(Kreidenweis and Seinfeld, 1988; Kreidenweis et al., 1989; Wyslouzil et al., 1991b, a; Dawson et al., 2012; Chen et al., 2015b), ammonia (Korhonen et al., 1999; Benson et al., 2009; Kirkby et al., 2011; Zollner et al., 2012), amines (Kurtén et al., 2008; Smith et al., 2010; Berndt et al., 2010; Erupe et al., 2011; Zollner et al., 2012; Yu et al., 2012; Almeida et al., 2013; Glasoe et al., 2015), and highly oxidized organic species (Zhang et al., 2004; Riipinen et al., 2011; Ehn et al., 2012; Zhao et al., 2013; Donahue et al., 2013; Kulmala et al.,

2013; Riccobono et al., 2014) also play a role. Many observations of NPF have been performed on the ground (Kulmala et al., 2004), but some measurements suggest that nucleation can be altitude dependent (Weber et al., 1999; Lee et al., 2003). Colder temperatures and higher relative humidity (RH) at higher altitudes may favor the binary nucleation of $H_2SO_4$ and $H_2O$, but at lower altitudes bases can play a substantial role due to the closer proximity to surface sources of these precursors (Weber et al., 1999). Ion-induced particle

formation is suggested to be important in the upper troposphere and lower stratosphere due to the stronger intergalactic cosmic rays that generate ions in these regions (Lee et al., 2003; Dunne et al., 2016). Vertically resolved measurements of NPF are therefore needed to better understand the species involved in NPF and the underlying mechanisms, as well as adequately assess the impacts of NPF on cloud formation and climate.

Only a few studies have focused on the vertical extent of new particle formation. During the SATURN

("Strahlung, vertikaler Austausch, Turbulenz und Partikel-Neubildung", or "Radiation, Vertical Exchange, Turbulence and New Particle Formation") campaign in Germany, tethered balloon-borne measurements were performed to obtain vertical profiles of $SO_2$ concentration, particle number concentration, and meteorological parameters together with ground-based measurements (Stratmann et al., 2003; Siebert et al., 2004; Wehner et al., 2007). The group observed two NPF events on the same day. The first one was initiated

in the residual layer before the breakup of the thermal inversion layer, and those newly formed particles were then mixed down during the subsequent breakup process. The second NPF event occurred in the entire well mixed layer during and after the break-up process of the thermal inversion layer. New particle formation can also be initiated near the top of the inversion layer due to intensive mixing caused by plumes penetrating the



inversion (Siebert et al., 2004), and within a thin layer around 200 m above the ground (Wehner et al., 2007). Laakso et al. (2007) also observed NPF in the mixed boundary layer and free troposphere in southern Finland using measurements on board a hot-air balloon.

The ability for a nanoparticle formed by NPF to grow to CCN size is dependent on its growth rate in relation to the rate of loss due to coagulational scavenging (McMurry and Friedlander, 1979; Kerminen and Kulmala, 2002; Kuang et al., 2009; Westervelt et al., 2013; Pierce et al., 2014). There has been much attention paid to understanding growth rates of newly formed particles. Thus far, mechanisms of new particle growth such as salt formation (Barsanti et al., 2009; Smith et al., 2010; Hodshire et al., 2016), condensation of extremely low-volatility organic compounds (ELVOCs) (Riipinen et al., 2011; Donahue et al., 2011; Riipinen et al., 2012; Jokinen et al., 2012; Ehn et al., 2014; Hodshire et al., 2016; Pierce et al., 2011), and reactive uptake of organics through particle phase reactions such as aldol condensation (Jang et al., 2002; Limbeck et al., 2003; Barsanti and Pankow, 2004), have been identified. Many of these reactions in newly formed particles might be enhanced by high water content, which has been shown for submicron particles in many field, laboratory and modelling studies (Blando and Turpin, 2000; Hennigan et al., 2008; Galloway et al., 2009; Ervens et al., 2011; Carlton and Turpin, 2013). In addition, the enhancing effect of water in NPF and growth was observed in the reaction of methanesulfonic acid and amines (Chen et al., 2015a; Chen et al., 2015b). While many field and laboratory measurements over the past several decades have quantified water uptake of submicron size particles, very little is known about the water content of newly formed particles, especially how it responds to relative humidity that is vertically and temporally variable in the atmosphere.

This manuscript describes vertical profile measurements of 11-16 nm diameter atmospheric aerosol particles from a tethered balloon during the New Particle Formation Study at the Department of Energy (DOE) Atmospheric Radiation Measurement (ARM) Southern Great Plains (SGP) research site in Lamont, Oklahoma. Those observations were supplemented by simultaneous ground-based measurements of particle number size distributions and size-resolved particle hygroscopicity. We explore the use of Raman Lidar measurements of vertically and temporally resolved temperature and relative humidity to provide insights into the range of liquid water content for newly formed particles. The effect of liquid water content of newly formed particles on particle growth is examined by a growth model. The goals of this study are to (1) fully characterize the time and location of new-particle formation events at this site, and (2) provide insights into the liquid water content of newly formed particles and their potential to undergo aqueous reactions and/or activate into cloud droplets.





## 2. Methods

### 2.1. Measurements

Vertical profiles of 11-16 nm diameter particles were measured as part of the 2013 SGP New Particle Formation Study (NPFS) (Hodshire et al., 2016). NPFS was sponsored by the U.S. Department of Energy

with participation from the University of Delaware, University of Minnesota, Augsburg College, and the National Center for Atmospheric Research (NCAR). This field study addressed the formation and growth of particles as well as the impacts of newly formed particles on cloud processes. NPFS took place at the Central Facility of the SGP site, spanning the dates of April 13-May 24, 2013. The site is representative of the large Great Plains region, and characterized by active agricultural activities, as well as oil extraction and refineries.

During the study, 13 NPF events were observed. Guest instruments that are relevant to the current study include three ground-based scanning mobility particle sizers (SMPSs) to measure the particle size distribution from 1.9-528 nm, and two hand-held condensation particle counters (CPCs) for particle concentration measurements deployed on a tethered balloon.

Three SMPSs were operated in parallel on the ground, including a DEG SMPS (a TSI 3085 nano-differential

mobility analyzer coupled with a diethylene glycol-based condensation particle counter; 1.9-13.9 nm mobility diameter) (Jiang et al., 2011), a nano SMPS (a TSI 3085 nano-differential mobility analyzer coupled with a TSI 3025A particle counter; 2.8-47 nm mobility diameter) and a conventional SMPS (a home-built long column differential mobility analyzer, similar in design to a TSI 3071, coupled with a TSI 3760 CPC; 23-528 nm mobility diameter). While measurements overlapped, the data were merged to provide the best

estimate of the correct size distribution.

The tethered balloon was a helium-filled blimp-shaped balloon made by Blimpworks (Statesvill, NC, USA) owned and operated by NCAR (Greenberg et al., 1999; Greenberg et al., 2014). The balloon was 6.4 m long, ~1.5 m in diameter at the widest point, and had a filled volume of 12 m$^3$. The height of the balloon was controlled by an NCAR-built winch with circuitry that provides a user-defined constant rate of altitude

change. Two hand-held condensation particle counters (CPCs; TSI 3007) were mounted under the balloon, each adjusted to a different diameter cut-off point. This enabled the ability to count particles between 11 and 16 nm by taking the difference in the two measurements. A small weather meter was used to log altitude (pressure), temperature and wind speed. Details of this payload are provided below.

The CPCs allow detection of particles by exposing the sampled aerosol to a warm saturated alcohol vapor.

This flow is then cooled using a thermal electric device (TED), causing the alcohol vapor to condense onto particles and grow particles to micron-sized droplets that can be detected by light scattering. The voltage applied to the TED directly affects the temperature difference between the saturated and condenser regions





within the CPC, which in turn defines size-dependent detection efficiencies as shown in Figure 1 for TED voltages of 1600 and 1000 mV. The number concentration of particles in the size range of 11-16 nm ($N_{11-16}$) was estimated by subtracting measurements from the CPCs with TEDs operating at 1600 and 1000 mV. The time resolution of CPC measurements is one second.

A battery-powered hand-held meteorological gauge (Kestrel 4500, Nielsen-Kellerman Co.) was attached to the tethered balloon, approximately 20 cm above the CPCs, which were ~0.5 m below the bottom of the balloon. Since the balloon always points into the wind, the wind direction measurements were obtained by the compass feature. Other meteorological parameters such as altitude, wind speed, RH, and temperature were also recorded by the gauge.

Launches were made during daylight hours. The balloon was raised and lowered at a constant rate of 0.5 m s$^{-1}$, and reached a maximum altitude of 800-1200 m above ground level. One launch, defined as one cycle of raising and lowering the balloon took place over a minimum time of ~46 min. During periods of continuous sampling, we repeated launches every ~60 min, which allowed for a brief break period for the operators to replace batteries, assess data quality, and perform any other required maintenance. Four days were studied
with the tethered balloon: May 12, 13, 16, and 17, 2013. The day of May 12, 2013 corresponded to a NPF event. The day of May13, 2013 was an intermittent NPF day, on which the growth of new particles was interrupted by the change of wind direction. The other two days were non-event days, and are not discussed in this manuscript.

In addition to guest instruments provided by NPFS participants, the SGP site was equipped with a variety of
instruments for probing atmospheric structure and properties, as well as the physicochemical properties of traces gases and particles. Instruments relevant to the current study include a Tandem Differential Mobility Analyzer (TDMA) (Collins, 2010) to measure size-resolved hygroscopicity of particles, Doppler Lidar (DL) (Newsom, 2012) to measure vertical wind velocity, and Raman Lidar (RL) (Turner, 2009) to measure vertical profiles of RH and temperature. The planetary boundary layer (PBL) heights were obtained from the ARM
value-added product that determines PBL height with radiosonde measurements based on the profile of the bulk Richardson number (Sivaraman et al., 2013). The radiosonde measurements were performed at 05:00, 11:00, 17:00, 23:00 UTC (00:00, 06:00, 12:00, 18:00 local time) each day, resulting in four PBL heights at their corresponding times. Linear interpolation of two consecutive measurements was used to estimate the evolution of PBL.





### 2.2. Estimates of hygroscopicity parameter, water content of nanoparticles, and atmospheric stability

The TDMA system consists of two high-flow differential mobility analyzers (Aerosol Dynamics, Inc.), a charger, two Nafion tubes and a CPC (Collins, 2010). The air sample was first dried in a Nafion tube, and

then drawn through a charger before entering the first differential mobility analyzer. The voltage applied to the first differential mobility analyzer was fixed to produce a mono-mobility aerosol of known particle mobility diameter. The monodisperse aerosol was then introduced into the second Nafion tube to elevate RH to 90%, and the growth of the monodisperse aerosol was characterized by the second differential mobility analyzer with a CPC. A typical measurement sequence required roughly 45 minutes. The TDMA was

calibrated every night shortly after midnight by injecting a polydispersed, pure ammonium sulfate aerosol that has a known hygroscopicity.

The TDMA system provides size-resolved normalized distribution of hygroscopic growth factor ($f_i$), which is defined as the ratio of the hydrated particle diameter at RH 90% to the dry diameter, throughout a day. The hygroscopicity parameter $\kappa$ for a certain size of particles at the corresponding time of a day can be calculated

by (Kreidenweis et al., 2008; Petters and Kreidenweis, 2007),

$$\kappa = \sum \varepsilon_i \kappa_i = \sum \varepsilon_i \left( f_i^3 - 1 \right) \left[ \left( \exp\left( \frac{4\sigma_{s/a}M_w}{RT\rho_w D_{dry}f_i} \right) / RH \right) - 1 \right] \qquad \text{Eq. (1)}$$

where $\varepsilon_i$ is the normalized fraction of aerosol with a hygroscopic diameter growth factor ($f_i$), $\sigma_{s/a}$ is the surface tension of the solution/air interface and is assumed to be the same as the surface tension of water, 0.072 J m$^{-2}$, $M_w$ is the molecular weight of water, $R$ is the universal gas constant, $T$ is the temperature during

the TDMA measurements and is assumed to be 298.15 K, $\rho_w$ is the density of water, $D_{dry}$ is the diameter of the dry particles, and $RH$ is the relative humidity (0.9).

Since RH and temperature profiles are available from Raman Lidar measurements, we combine these data with measurements of TDMA-derived measurements of $\kappa$ to estimate the profile of water content of newly formed particles. The growth factor ($f$) and water volume ratio ($\phi$) of particles at elevated heights within the

boundary layer can be estimated using Equations (2) and (3) (Petters and Kreidenweis, 2007; Kreidenweis et al., 2008) assuming that the boundary layer is well mixed so that particles at elevated heights have the same $\kappa$ as particles measured at the ground level,

$$RH = \frac{f^3 - 1}{f^3 - (1 - \kappa)} \exp\left( \frac{4\sigma_{s/a}M_w}{RT\rho_w D_{dry}f} \right) \qquad \text{Eq. (2)}$$

$$\phi = \frac{V_{wet}}{V_{dry}} = f^3 \qquad \text{Eq. (3)}$$





where $T$ and $RH$ are the temperature and relative humidity at selected time and height obtained from Raman Lidar measurements, $V_{wet}$ is the volume of the wet particle, and $V_{dry}$ is the volume of the dry particle. The growth factor $f$ was obtained by solving the cubic Equation (2) in the physically meaningful range of $f > 1$. Supersaturation of more than 1–2% relative to water is rarely seen in the atmosphere due to the prevalent

existence of CCN (Rogers and Yau, 1989). Therefore, points with RH > 101% were treated as systematic errors and neglected from analysis. When RH is between 100-101%, there are two solutions to Equation (2), but only the lower one is at a stable state. When RH is lower than 100%, there is only one solution of $f$ for $f > 1$ (Seinfeld and Pandis, 2006; Petters and Kreidenweis, 2007).

While the surface tension of water, 0.072 J m$^{-2}$, was used in Equations (1) and (2), atmospheric particles are

known to contain some surface active compounds (Facchini et al., 1999; Svenningsson et al., 2006). The presence of surface active compounds in particles can suppress surface tension to a significant extent. Therefore, sensitivity studies were carried out by decreasing $\sigma_{s/a}$ used in Equations (1) and (2) to examine its effects on estimated $\kappa$ values and water volume ratios of particles.

In order to determine the stability of the boundary layer, the Richardson number ($R_i$) was calculated using

Equation (4) (Woodward, 2010),

$$R_i(z_1) = \frac{g}{T(z_1)} \frac{\left[\frac{T(z_1)-T(z_2)}{z_1-z_2}\right]}{\left[\frac{u(z_1)-u(z_2)}{z_1-z_2}\right]^2} \qquad \text{Eq. (4)}$$

where $z$ is the height, $g$ is the gravitational acceleration, $T$ is the ambient temperature at the reference height obtained from Raman Lidar measurements, and $u$ is the vertical wind speed at the reference height obtained from Doppler Lidar measurements. Based on the Richardson number, the stability of the boundary layer can

be classified into six Pasquill stability regimes: A - extremely unstable; B - moderately unstable; C - slightly unstable; D - neutral; E - slightly stable; and F - moderately stable.

The atmospheric stability within the boundary layer on May 12, 2013 during the campaign is shown in Figure 2. Data herein are provided in UTC time, which is 5 hours ahead of the local time during the campaign. Figures showing the atmospheric stability of other selected days of interest are included in Figure S1 in the

supplemental information. The boundary layer was moderately stable at night, suggesting that a thermal inversion layer developed at night and suppressed convection. The boundary layer was under extremely or moderately unstable conditions during the daytime because the thermal inversion layer was dissipated and eventually disappeared after sunrise. Given that NPF events in this study occurred during the daytime, a well mixed boundary layer is a reasonable assumption. While a well mixed boundary layer makes it likely that

particles measured at ground level are representative of those aloft, it does not guarantee it. This should be borne in mind in the subsequent discussion.





### 2.3. Model description

In order to explore the effect of increasing relative humidity on the relative uptake of each chemical species into particles, the particle growth model MABNAG (Model for Acid-Base chemistry in NAnoparticle Growth) is used to model the NPF events observed on the case days of April 19, May 9, and May 11 over a

relative humidity range of 30-95% in 5% increments. These days were selected because extensive measurements of particle composition and gas-phase species were performed; these observations were presented in detail in Hodshire et al. (2016). Each day featured with different dominant species contributing to particle growth: April 19 was from ELVOCs, May 9 was from inorganic salts like ammonium sulfate, and May 11 was from ELVOCs, amines/ammonia, and sulfate. MABNAG, developed by Yli-Juuti et al. (2013),

simulates the growth and composition of a single particle as it undergoes condensation of low-volatility vapors and acid-base reactions in the particle phase. The version of MABNAG used in this study accepts as inputs the gas-phase concentrations and properties (molar mass, pKa, vapor pressure, and Henry's Law constant or diffusion coefficient) of water, sulfuric acid, a single organic acid (to be representative of organic acids in general), ammonia, a single amine (chosen to be representative of amines in general), and a

nonreactive organic compound, assumed here to be an ELVOC. Malonic acid and dimethylamine (DMA) were selected as the representative organic acid and base, respectively in this study. The initial particle is assumed to be made of 20 molecules of each input species, creating a particle that is approximately 3 nm in diameter. The particle density is assumed to be species-independent and is set to 1.5 g cm$^{-3}$; the particle surface tension is set to 0.03 J m$^{-2}$. MABNAG calculates the uptake rates of sulfuric acid, the organic acid,

and the ELVOC species as gas-phase diffusion-limited mass transfer as functions of their ambient vapor pressures, equilibrium vapor pressures, and gas-phase diffusivities. Water and the base species are assumed to reach instantaneous equilibrium between gas and particle phases. Upon uptake, particle-phase acid dissociation and base protonation are calculated by the Extended Aerosol Inorganics Model (E-AIM) (http://www.aim.env.uea.ac.uk/aim/aim.php; Clegg et al., 1992; Wexler and Clegg, 2002; Clegg and

Seinfeld, 2006b, a). The ELVOC is not allowed to dissociate in the particle phase, and is assumed to undergo irreversible condensation due to its low vapor pressure.

As discussed in Hodshire et al. (2016), there are limitations and uncertainties to many of the MABNAG inputs. MABNAG can only accept the chemical properties of one organic acid and one amine, although both oxalic and malonic acid were measured and many different amines were observed. Further, there are

uncertainties in the measured oxalic acid concentration, the vapor pressures of the organic acids, and whether or not the larger amines (containing at least four carbons) participate in particle growth. For simplicity, we model only the base case from Hodshire et al. (2016) over the given relative humidity range (30-95%). This base case assumes the organic acid properties of malonic acid, the equilibrium vapor pressure for malonic



acid to be that as reported in Bilde et al. (2015), the organic acid concentration to be that of the sum of the measured malonic acid and 10 times the measured oxalic acid concentration (due to a likely low bias of the oxalic acid measurements), the amine chemical properties of DMA, and the amine concentration to be the sum of the measured methylamine, DMA, and trimethylamine only. A more-detailed description of the

inputs used for this study and the conditions of the three case days can be found in Hodshire et al. (2016).

## 3. Results and discussion

### 3.1. Vertical profiles of particle number concentration in the size range of 11-16 nm in relation to ground-based measurements of number-size distribution

Figure 3 shows the NPF event on May 12, 2013 at the ARM SGP site as captured by the ground-based SMPS

and tethered balloon system. A burst of small particles appeared approximately at 15:00 UTC (10:00 local time), and they continually grew to ~30 nm within the next several hours (Fig. 3a). The tethered balloon was launched three times on this day, and the corresponding vertical profiles of particle number concentration in the size range of 11-16 nm ($N_{11-16}$) are shown in Figures 3b-d. The vertical profiles of RH and temperature are shown in Figure S2 in the supplemental information. For the first ascent starting at 14:48 UTC (Fig. 3b),

$N_{11-16}$ is negligible within the lowest 100 m, and increases gradually as the balloon raised up to ~100 m. Maximum values of up to $2.0 \times 10^4$ cm$^{-3}$ were observed at 300-400 m. $N_{11-16}$ is almost zero as the balloon reached up to ~400 m. During the subsequent descent, $N_{11-16}$ is relatively constant at ~$1.0 \times 10^4$ cm$^{-3}$, and shows little variation with height. The vertical profile of temperature (Fig. S2a) shows a lapse rate between 0.9 and 1.2 ºC per 100 m up to ~400 m. The profile of relative humidity (Fig. S2a) features some fluctuations

between 45%-55% at the lowest 400 m but a rapid decrease at 400 m, at the same height where a small increase of temperature and a sharp decrease of $N_{11-16}$ were observed. These together indicate that during the first launch of the tethered balloon, the mixed layer had a height of about 400 m.

Compared to the first launch, the values of $N_{11-16}$ for the second and third launches are lower (Fig. 3c-d). From the second ascent starting at 19:42 UTC to the third descent ending at 21:48 UTC, $N_{11-16}$ keeps

decreasing with time, and does not exhibit any sharp fluctuation with height. The vertical profiles of temperature and RH for the two launches are similar, showing a temperature lapse rate between 0.9 and 1.0 ºC per100 m, and increasing RH with height.

To further compare ground-based and balloon-borne measurements, particle number concentrations in the size range of 11-16 nm measured by the ground-based SMPS during the first launch were integrated by

taking into account the slopes of the detection efficiency curves of the two balloon-borne CPCs. The result is shown in Figure 4. During the ascent, $N_{11-16}$ measured aloft is up to 10-fold larger than that measured by the





ground-based SMPS, while $N_{11-16}$ measured by the two systems agree reasonably well during the subsequent descent of the balloon. The ground-based SMPS together with the balloon-borne measurements suggest that new particle formation was initiated at the top region of the boundary layer, and subsequently mixed downward to ground level due to the unstable conditions and therefore the strong vertical mixing within the

boundary layer (Fig. 2).

Similar measurements were made during an interrupted NPF day of May 13, 2013 (Fig. 5). A burst of small particles appeared at 14:00 UTC at the ground level, and continually grew to ~10 nm before interrupted by the change of wind direction from southwest ($180^o$ - $220^o$) to northwest ($270^o$ - $310^o$) at about 16:00 UTC (Fig. 5a). The growth of newly formed particles was observed after the wind direction changed back to

northwest at about 19:00 UTC. Some shorter interruptions followed in the late afternoon (20:00-23:00 UTC). Six balloon launches were made on this day, and the corresponding vertical profiles of $N_{11-16}$ as well as temperature and RH are shown in Figures 5b-g and Figure S3, respectively. Since the initial period of the NPF event was not captured by the tethered balloon system, it is not known whether the NPF was initiated aloft or at the ground level. However, similar to the NPF event on May 12, the entire boundary layer appeared

to be well mixed with $N_{11-16}$ in about one hour after NPF was first observed on the ground.

The comparison of ground-based and balloon-borne measurements for the two days provides insights into the vertical extent of NPF and the vertical mixing process within the boundary layer. The NPF event on May 12 was initiated at the top region of the boundary layer followed by rapid downward mixing process. The timescale of the vertical mixing was approximately 0.5-1 hour. Although the background concentration of

particles before NPF was not captured by the tethered balloon system on May 13, the boundary layer appeared to be well mixed with ultrafine particles, suggesting a strong vertical mixing on this day.

### 3.2. Size-resolved hygroscopicity of particles

Size-resolved one-hour average values of the hygroscopicity parameter, $\kappa$, for the whole campaign are shown in Figure 6. The $\kappa$ values of 13-nm particles ($\kappa_{13nm}$) are in the range of 0.5-0.9, significantly larger than those

of larger-size particles, and exhibit a distinct diurnal pattern of increasing in the daytime and decreasing at night. The values of $\kappa_{25nm}$ are in the range of 0.1-0.3, much smaller than $\kappa_{13nm}$ but slightly larger than those of larger size particles. The $\kappa$ values of larger size particles are similar, and do not show a distinct diurnal pattern. While $\kappa$ values in Figure 6 were calculated assuming that the surface tension of solution/air interface is the same as water (0.072 J m$^{-2}$), a sensitivity analysis was carried out by decreasing $\sigma_{s/a}$ by 30% (0.0504 J

m$_{-2}$) or using $\sigma_{s/a}$ as the one used in the MABNAG model (0.03 J m$^{-2}$). The values of $\kappa_{13nm}$ are still distinctly larger than those of larger-size particles (Fig. S4).



The hygroscopicity of particles is strongly dependent on their chemical composition. Particles composed of inorganic salts, acids, and highly oxidized organic species grow significantly in response to increased RH, while particles containing less oxidized organic species, soot and dust exhibit little hygroscopic growth. Values of $\kappa$ are in the range of 0.5-1.4 for atmospherically relevant salts and small acids, and 0.01-0.5 for

slightly to very hygroscopic organic species (Petters and Kreidenweis, 2007). The growth factor of nanoparticles of the same chemical composition decreases with size due to the Kelvin effect (Hämeri et al., 2000; Hämeri et al., 2001; Park et al., 2009; Lewis, 2006). The higher $\kappa$ values of 13-nm particles observed at the SGP site suggest that they contained a larger fraction of highly hygroscopic species such as inorganic salts, acids, and highly oxidized organic compounds. Gaseous precursors of those highly hygroscopic species

might be formed photochemically (e.g. oxidation of $SO_2$ to form $H_2SO_4$, oxidation of volatile organic compounds to form ELVOCs), or might be enhanced due to higher emissions at elevated daytime temperatures (e.g. emission of $NH_3$ and amines from soil and animal husbandry), resulting in the distinct diurnal variation of $\kappa_{13nm}$. As particles grew larger, less oxidized organic species contributed to particle growth, leading to lower values of $\kappa$.

**3.3. Estimates of vertically resolved nanoparticle liquid water content**

While ultrafine particles are highly hygroscopic, and can be formed aloft as discussed earlier, the profiles of water volume ratio of 13-nm particles were estimated based on the RH and temperature profiles obtained from Raman Lidar measurements, assuming that $\kappa_{13nm}$ is representative of 13 nm particles throughout the boundary layer. This assumption is supported by the observation of the extremely unstable boundary layer

during the daytime based on the Richardson number analysis (Fig. 2, S1). The profiles of estimated water volume ratio for 13 nm particles on May 12 and May 13, 2013 are shown in Figure 7. Also shown in Figure 7 are temporally resolved $\kappa_{13nm}$ obtained by TDMA, and the RH (upper plots) and temperature (white contour lines in the lower plot) profiles of ambient air on these days. Consistent with the tethered balloon measurements (Fig. S2, S3), relative humidity is in the range of 30-60%, and is higher at the top of the

boundary layer after 18:00 UTC. The increased RH is possibly due to increasing boundary layer depth and lower temperature at higher altitude (Ek and Mahrt, 1994). Temperature decreases with increased height in accord with tethered balloon measurements. The $\kappa_{13nm}$ values show an increasing trend as NPF was initiated on May 12 and May 13, 2013, suggesting that highly hygroscopic species were involved in NPF and subsequent growth. The water volume ratios ($\phi$) of 13-nm particles are highly variable depending on RH,

temperature and $\kappa$, and were estimated to be in the range of 1-2.

It is noteworthy that $\sigma_{s/a}$ can be smaller than the surface tension of water, 0.072 J m$^{-2}$ if surface active compounds were present as discussed earlier. A sensitivity study was carried out by decreasing $\sigma_{s/a}$ used in





Equations (1) and (2) to 0.0504 and 0.03 J m$^{-2}$ to examine its effects on $\kappa$ values and estimated water volume ratios of particles. Using 30% and 58% lower values of $\sigma_{s/a}$ results in ~17% and ~30% lower values of $\kappa$, respectively (Fig. S5). However, these lower $\kappa$ was applied in Equation (2) along with the lower $\sigma_{s/a}$ to predict particle growth factor. Since $\kappa$ and $\sigma_{s/a}$ were applied self-consistently in Equations (1) and (2), water

volume ratios of particles estimated with 30% and 58% lower values of $\sigma_{s/a}$ are only ~2% and ~4% higher, respectively, suggesting that uncertainties in $\sigma_{s/a}$ have a relative minor effect in estimate of water volume ratio of particles.

There are other uncertainties in estimating the water content of 13-nm particles. First, although the boundary layer appeared to be well mixed based on the Richardson number, hygroscopicity parameter $\kappa$ of 13-nm

particles measured at the ground level might be not representative of $\kappa$ throughout the entire boundary layer. Figure S6 shows how water volume ratio of 13-nm particles varies with Kappa and RH at 298.15 K. While $\kappa_{13nm}$ was not measured at elevated heights, the variation of $\kappa_{13nm}$ measured at the ground, 0.5-0.9 (Fig. 6), is used to evaluate uncertainties of estimated water ratio here. Decreasing $\kappa_{13nm}$ from 0.9 to 0.5 results in a decrease of water volume ratio of 13-nm particles from 1.53 to 1.29 at RH 40%, and from 2.16 to 1.63 at RH

60%. This suggests that even if $\kappa_{13nm}$ at elevated heights is relatively low, particles can still have significant water content in ambient air. Second, $\kappa$ is defined through its effect on the water activity of the solution. While $\kappa$ was assumed to be independent of RH, the change of water activity with RH can lead to a non-constant $\kappa$ value over the range of RH, but the effect was suggested to be minimal (Petters and Kreidenweis, 2007).

In addition to the two days during which tethered balloon measurements were carried out, another three NPF events that featured three distinct growth pathways, and were discussed by Hodshire et al. (2016), were selected to estimate the profiles of water volume ratio. Figure 8 shows observations from May 9, 2013, during which particle growth was mainly driven by sulfuric acid and ammonia, with small but nontrivial contributions from organics and amines (Hodshire et al., 2016). The $\kappa_{13nm}$ value increases significantly to up

to ~1 as NPF started. The high value of $\kappa_{13nm}$ is consistent with sulfuric acid and ammonia as the main species contributing to NPF and growth. Relative humidity is in the range of 60-101%, and is significantly higher than that for the two days with tethered balloon measurements (Fig. 7). Because of the high $\kappa_{13nm}$ and RH, estimated water volume ratios of 13-nm particles are substantially higher, and can reach to ~18 at the top of the boundary layer.

For the day of April 19, 2013 (Fig. S7), on which particles grew mainly by acid-base reactions of organic acids and amines and/or irreversible condensation of ELVOCs, the $\kappa_{13nm}$ value stays constant at ~0.6 in the period of NPF and growth, and estimated water volume ratios of 13-nm particles are in the range of 1-3. For



the day of May 11, 2013 (Fig. S8), on which sulfate, organics, and bases are important for growth, an increase of $\kappa_{13nm}$ was observed as NPF started. The estimated water volume ratios of 13-nm particles during the day are in the range of 1-3. Our estimates of water volume ratios of 13-nm particles for five different NPF days suggest that newly formed particles were highly hygroscopic at the SGP site, and contained up to ~18-fold

volume increase in water compared to dry particles depending on chemical composition of particles and metrological conditions.

## 4.  Atmospheric implications

Particle liquid water content influences many atmospheric processes. For example, water in particles facilitates the partitioning of water-soluble gas species into the condensed phase (Blando and Turpin, 2000;

Surratt et al., 2006; Ervens et al., 2008; Hennigan et al., 2008; Galloway et al., 2009; Asa-Awuku et al., 2010; Carlton and Turpin, 2013; Hodas et al., 2014). In this study, the growth of nanoparticles was simulated with the MABNAG model (Yli-Juuti et al., 2013; Hodshire et al., 2016), in which RH was varied in the range of 30-95% while concentrations of other gas species were kept constant (Fig. 9). Three new particle formation and growth events simulated with the MABNAG model featured three distinct growth pathways. In each

case, increased amount of water in nanoparticles at higher RH enhances the partitioning of $NH_3$, DMA, and organic acids into the condensed phase, but has a negligible effect on the uptake of sulfuric acid and ELVOC, which already have extremely low saturation vapor pressures. Compared to the case of RH 30%, particle growth by organic acids is enhanced by a factor of 35-100 at RH 95%, and surpasses growth contributed by sulfuric acid and/or ELVOC in some cases. The modelling results indicate that even for nanometer-sized

particles, liquid water content significantly influences the gas-particle partitioning, and therefore chemical composition of particles. Gas species with extremely low saturation vapor pressures dominantly contribute to particle growth at low RH, but water-soluble gas species can be more important to particle growth as RH increases.

Water-soluble gas species (e.g. alcohols, aldehydes, ketones, acids, hydroperoxides, ammonia, amines) also

react within particles to form lower volatility species via reactions such as hemiacetal/acetal formation, aldol condensation, salt formation, and esterification (Ervens et al., 2011), though only salt formation is considered in the MABNAG simulations here. The increased partitioning of water-soluble gas species in ultrafine particles may substantially increase the growth of particles and yield of secondary organic aerosol (SOA) through these mechanisms. The partitioning of water-soluble oxidizing species like hydroperoxides in

ultrafine particles also increases the oxidizing capacity of particles, and enhances aqueous-phase oxidation processes, impacting SOA formation. In addition, it is well known that $H_2O_2$ causes significant damage to human lung cells (Gurtner et al., 1987; Oosting et al., 1990; Holm et al., 1991; Sporn et al., 1992). Ultrafine




particles associated with increased $H_2O_2$ can penetrate deeply into the lung, and impair lung function to a greater extent. Therefore, considering liquid water content in ultrafine particles in modeling studies is of great importance for accurately estimating particle growth, atmospheric chemical reaction rates, SOA formation and the effects of particles on human health.

## 5. Conclusions

In summary, we measured vertical profiles of particle number concentrations in the size range of 11-16 nm with a tethered balloon system at the U.S. Department of Energy ARM SGP field site in Oklahoma. These observations were compared with simultaneous ground-based data obtained from three scanning mobility particle sizers to understand the vertical extent of new particle formation. We found that new particle formation was initiated at the top region of the boundary layer, and subsequently mixed downward to ground level in 0.5-1 hour due to the strong vertical mixing on the NPF event day of May 12, 2013. The ground-based TDMA data indicate that newly formed particles at the SGP site are highly hygroscopic, and typically contain up to 50% water by volume depending on chemical composition of particles and metrological conditions. During conditions of high RH, liquid water may account for up to 95% of particle volume for newly formed 13 nm particles. The MABNAG simulation supports that relative humidity, which is vertically and temporally variable, has a substantial effect on the partitioning of gas species into nanoparticles. Specifically, the uptake of organic acids to nanoparticles is greatly enhanced at high RH, and can dominantly contribute to nanoparticle growth.

While field measurements are commonly performed at ground level, our results highlight the importance for direct measurements of vertically resolved properties of gas species and particles to better understand NPF and growth as well as its impacts on cloud processes and climate. In particular, measurements of the poorly understood water content of nanoparticles would greatly improve our understanding and modelling studies in gas-particle partitioning and growth of nanoparticles into submicron size particles that can serve as CCN.

**Acknowledgements**

This research was supported by the U.S. Department of Energy's Atmospheric System Research, an Office of Science, Office of Biological and Environmental Research program, under Grant No. DE-SC0011780 and Grant No. DE-SC0014469. We acknowledge the cooperation of the US Department of Energy as part of the Atmospheric Radiation Measurement (ARM) Climate Research Facility in hosting the NPFS at the Central Facility. We would also like to acknowledge the support of ARM staff at the SGP site, and Professor Don Collins from Texas A & M University for useful discussions regarding the TDMA data.





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



**Figures**

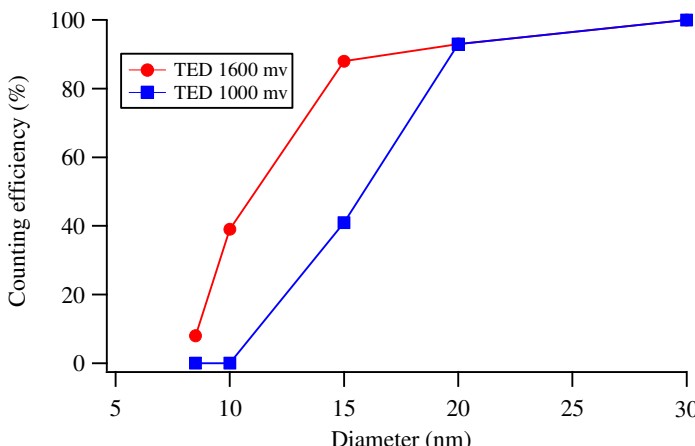

**Figure 1.** Counting efficiency as a function of particle diameter for two TSI 3007 CPCs while thermal
5  electric devices (TEDs) of the two CPCs were set at 1000 and 1600 mV, respectively. The counting
efficiencies were calculated relative to counts with thermal electric device set at 2000 mV for atomized
ammonium sulfate particles.





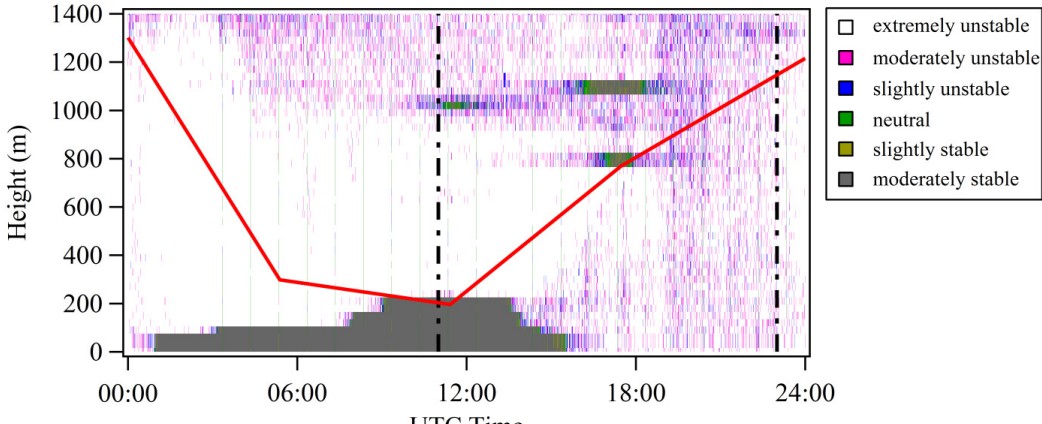

**Figure 2.** Vertically and temporally resolved atmospheric stability on May 12, 2013 at SGP. The atmospheric stablity is classified into extremely unstable (grey), moderately unstable (magenta), slightly unstable (blue), neutral (green), slightly stable (brown) and moderately stable (white) regimes based on Richardson number. The two vertical dash lines represent 06:00 and 18:00 local time, respectively. The red line shows the boundary layer height determined from radiosonde data.



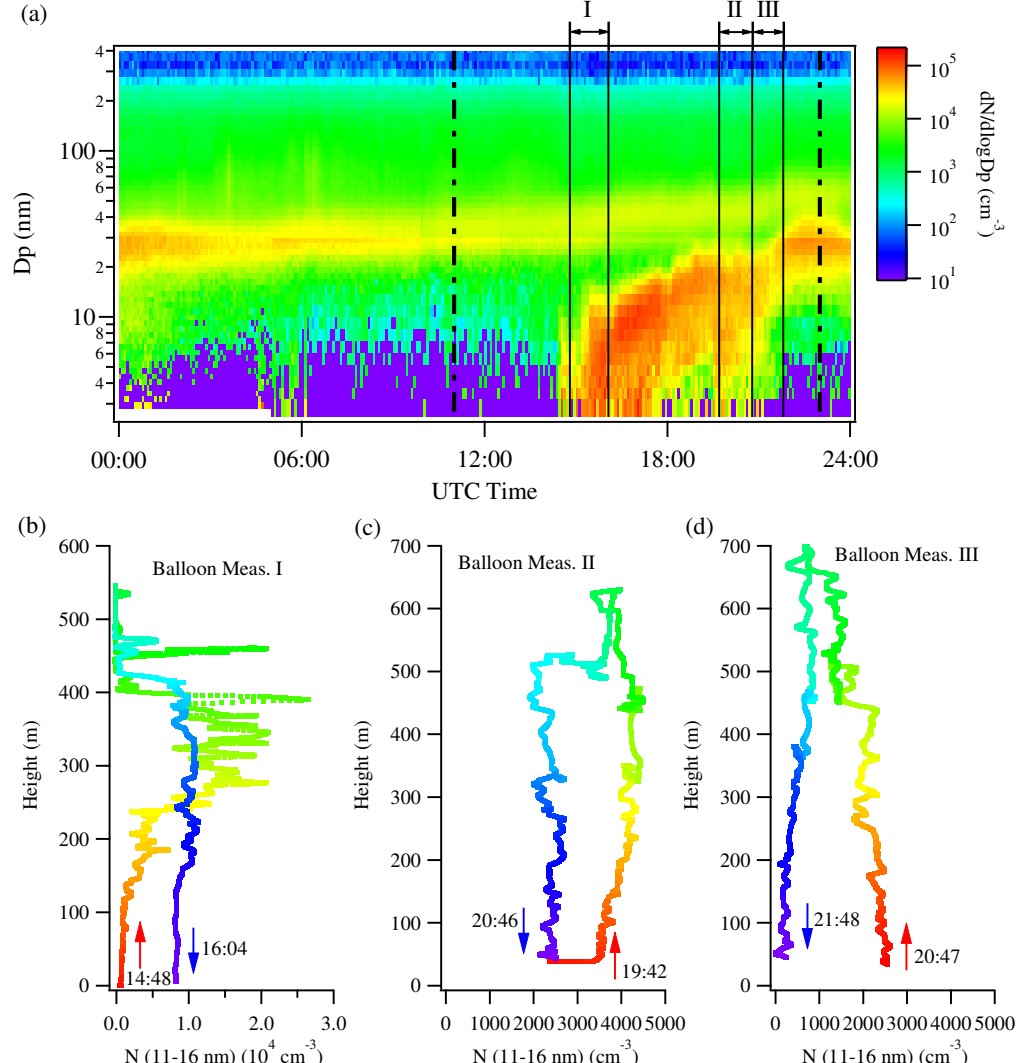

**Figure 3**. New particle formation event on May 12, 2013 at SGP. (a) Particle number concentrations captured by ground-based scanning mobility particle sizers. The two vertical dash lines represent 06:00 and 18:00 local time, respectively. The vertical black solid lines represent the time frames of the tetherd balloon measurements. (b-d) Vertical profiles of particle number concentration in the size range of 11-16 nm measured by the tethered balloon system. Three launches were made during the new particle formation event. A rainbow color pattern is used to illustrate the time after the balloon measurements were started, with red representing the initial period and purple representing the final period of a launch.



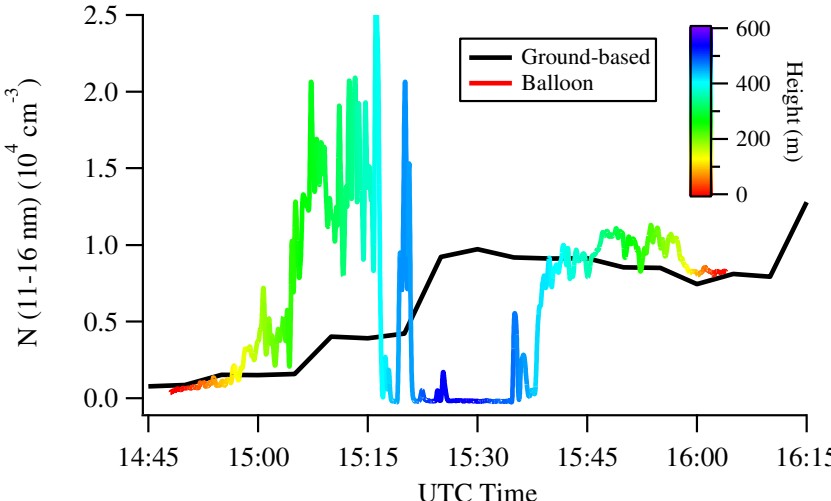

**Figure 4.** Time evolution of particle number concentrations in the size range of 11-16 nm at SGP during the intial period of new particle formation on May 12, 2013 obtained by the ground-based ground-based scanning mobility particle sizers and tethered balloon system.



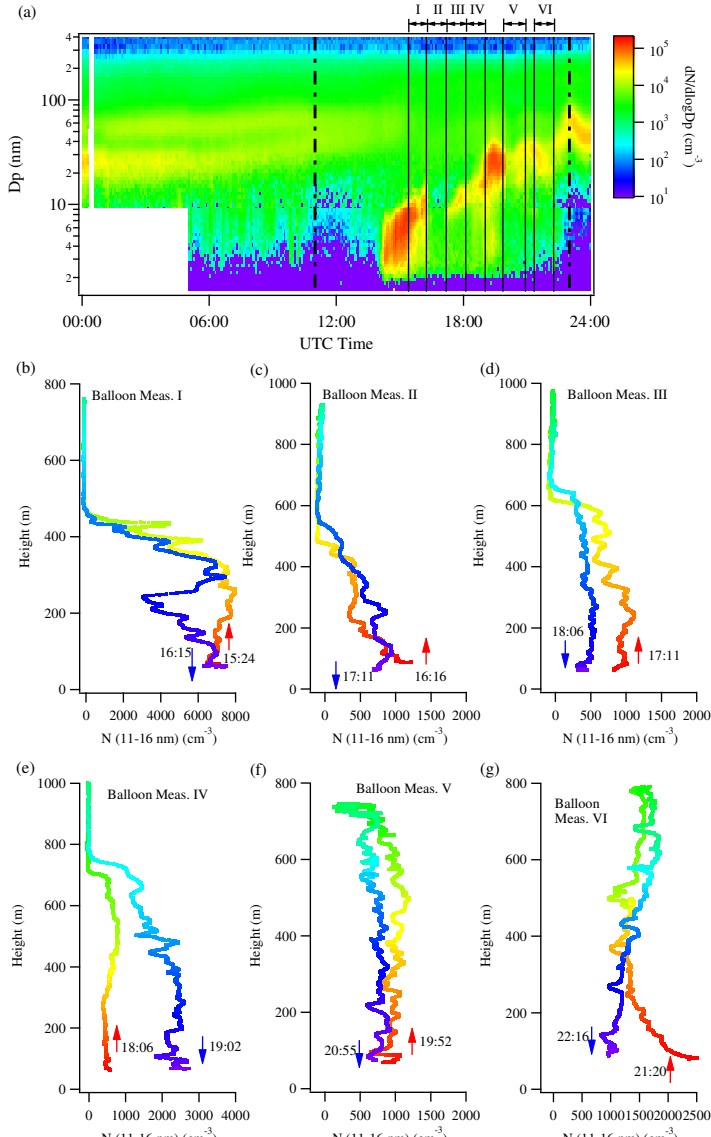

**Figure 5**. New particle formation event on May 13, 2013 at SGP. (a) Particle number concentrations captured by ground-based scanning mobility particle sizers. The two vertical dash lines represent 06:00 and 18:00 local time, respectively. The vertical black solid lines represent the time frame of the tetherd balloon measurements. (b-g) Vertical profiles of particle number concentration in the size range of 11-16 nm measured by the tethered balloon system. Six launches were made during the new particle formation event. A rainbow color pattern is used to illustrate the time after the balloon measurements were started, with red representing the initial period and purple representing the final period of a launch.



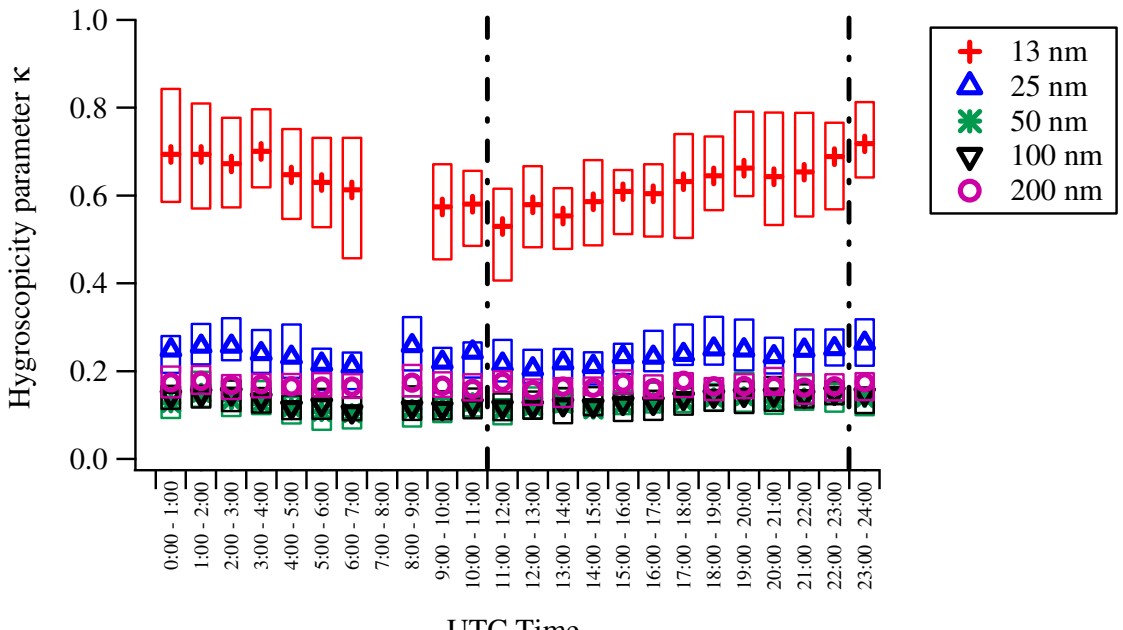

**Figure 6.** One-hour average hygroscopicity parameters $\kappa$ for 13, 25, 50, 100 and 200 nm particles determined by TDMA from April 13 - May 24, 2013 at SGP. The boxes represent 25-75% ranges of all measurements. The two vertical dash lines represent 06:00 and 18:00 local time, respectively.





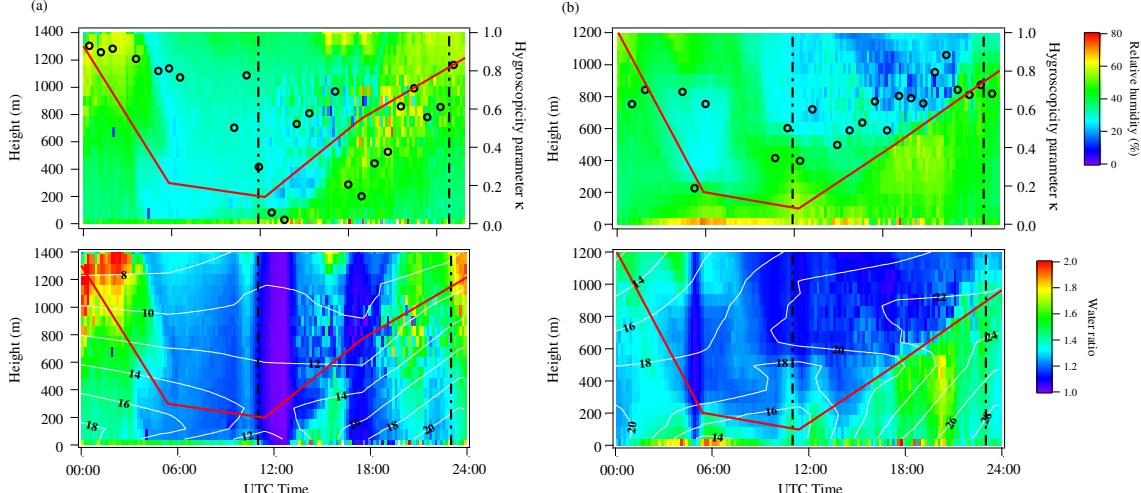

**Figure 7**. Vertically and temporally resolved relative humidity (upper plot) and estimated water volume ratio of 13-nm particles (lower plot) at SGP on (a) May 12, 2013, and (b) May 13, 2013. The two vertical dash lines in each plot represent 06:00 and 18:00 local time, respectively. The red lines shows the boundary layer height determined from radiosonde data. Overlaid in the upper plot is the temporally resolved hygroscopicity paramter $\kappa$ of 13-nm particles obtained by TDMA (open black circles). Temperature (°C) is shown with the white contour lines in the lower plot.



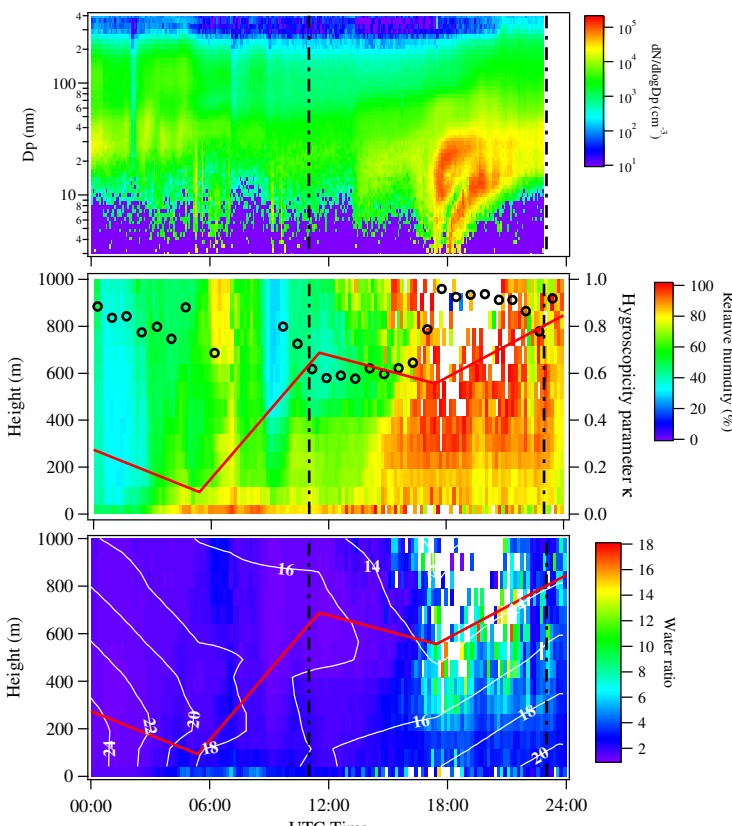

**Figure 8.** New particle formation event captured by scanning mobility particle sizers (upper plot), vertically and temporally resolved relative humidity (middle plot) and estimated water volume ratio of 13 nm particles (lower plot) on May 9, 2013 at SGP. The two vertical dash lines in each plot represent 06:00 and 18:00 local time, respectively. The red lines shows the boundary layer height determined from radiosonde data. Overlaid in the middle plot is the temporally resolved hygroscopicity paramter $\kappa$ of 13-nm particles obtained by TDMA (open black circles). Temperature (°C) is shown with the white contour lines in the lower plot.



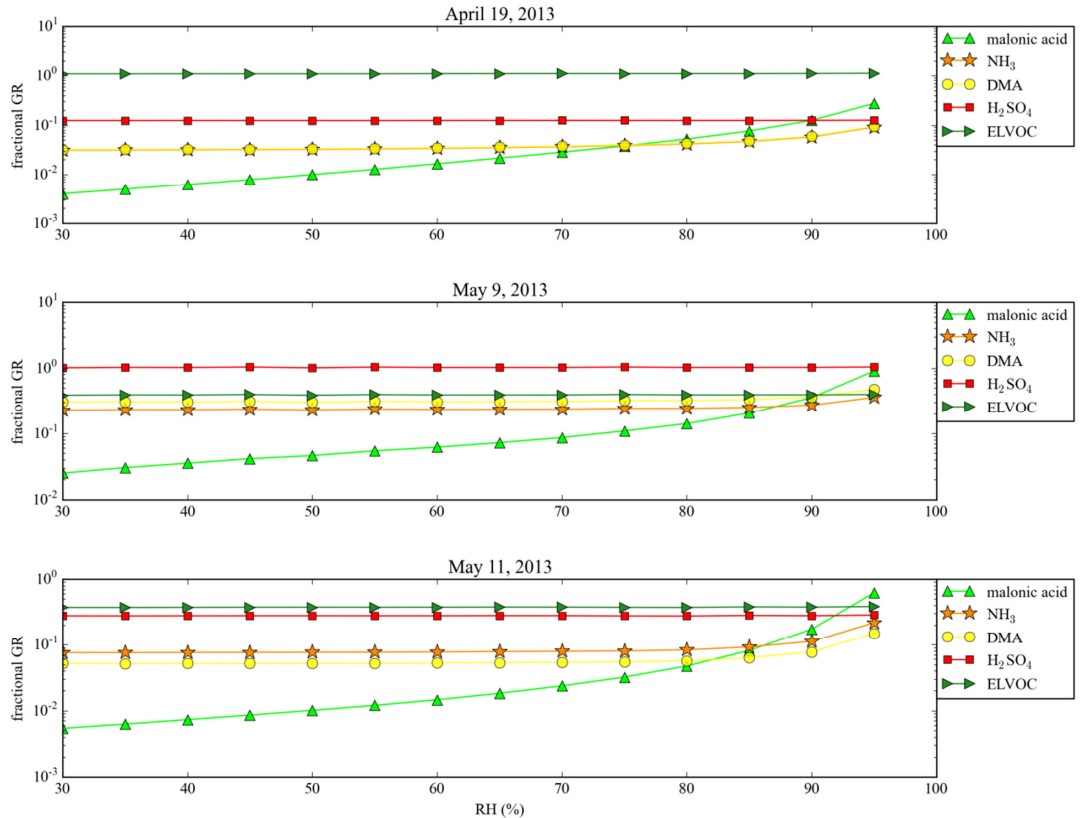

**Figure 9**. MABNAG prediction of growth rates of 13-nm particles contributed by individual species as a function of relative humidity for three new particle formation events at SGP.