# Peer review of "Vertically resolved concentration and liquid water content of atmospheric nanoparticles at the US DOE Southern Great Plains site"

_Atmospheric Chemistry and Physics, 2017_

## Referee Comment (RC1) · Anonymous Referee #1 · 10 Aug 2017

The manuscript presents important novel measurements of the concentration and liquid water content of atmospheric nanoparticles as a function of altitude. The authors do an excellent job of identifying and testing the assumptions in their analysis. Please consider the following comments:

(1) Page 2, bottom paragraph: It is interesting to consider the authors' results in light of a contemporaneous 2017 ACPD manuscript from Andreae et al., who observed NPF in the upper troposphere of the Amazon and condensational growth in the planetary boundary layer. The authors brought up earlier measurements from SATURN and southern Finland but missed helicopter/aircraft studies in the remote summertime Arc-

tic (e.g., Kupiszewski et al., ACP 2013; Burkart et al., ACP 2017) that found ultrafine particles near the surface, with much lower concentrations in the free troposphere.

(2) Page 4, lines 25-27: What were the explicit size ranges for each CPC? What is the uncertainty associated with the 11-16 nm size window?

(3) Figure 2: The grey and white colors in the legend are reversed.

(4) Captions for Figures 3 and 5: "Tethered" is misspelled in line 4.

(4) In section 3.1, how representative are the two days for general conditions at the site? How do they compare to the other days simulated in Figure 9?

---

## Referee Comment (RC2) · Anonymous Referee #2 · 26 Sep 2017

The paper presents an important topic, the vertical profile of new particle formation and further estimates and speculates on the LWC of these newly formed particles. The paper uses two CPCs of different lower cut-off size for the profile measurements of 11-16 nm particles. Further on the authors take advantage of the ground level SMPS and HTDMA measurements and a model. The paper is clearly written and the presentation of results is well performed explaining the assumptions needed and uncertainties in the analysis. Still, I recommend the authors to consider the following comments.

General Comments:

- Could you justify the selection of 11-16 nm range for the profile measurements?

[Figure]

- As the profiles are measured with 11-16 nm particles, the observed particles are the particles that have been growing 1-3 hours after the actual onset of the NPF event itself. Does this affect the analysis and what kind of uncertainty this brings in?

- Based on the fact above, as the 11-16 nm particles have been first observed at well above ground level, could it be that this is due to a larger growth rate from 1 nm to 11 nm at this altitude, and also more intense burst of particles?

Specific comments:

- Page 4, rows 3-20: Are the particles measured with the profile CPCs and ground-based SMPSs dry or wet/ambient?

- Page 6, row 3-11: Please mention the used HTDMA diameters already here.

- Page 10, row 24: How much did the hygroscopic growth factor and Kappa-value varied throughout the period and especially on the case days 12-13 May?

- Figure 2: The legend and figure caption colors do not match.

---

## Author Comment (AC1) · 3 Nov 2017

Vertically resolved concentration and liquid water content of atmospheric nanoparticles at the US DOE Southern Great Plains site

Responses to Comments of Reviewers 1 and 2

The authors are grateful for the careful review and helpful comments offered by both Reviewers, and we have revised our manuscript accordingly. Each comment is repeated below and our response follows with suggested changes to the manuscript given in quotes. We have also attached the modified manuscript (PDF format) with

insertions and deletions noted in colored font. Also note that two figures are supplied in this response.

Reviewers' Comments to Author:

Reviewer: 1

The manuscript presents important novel measurements of the concentration and liquid water content of atmospheric nanoparticles as a function of altitude. The authors do an excellent job of identifying and testing the assumptions in their analysis. Please consider the following comments:

1. Page 2, bottom paragraph: It is interesting to consider the authors' results in light of a contemporaneous 2017 ACPD manuscript from Andreae et al., who observed NPF in the upper troposphere of the Amazon and condensational growth in the planetary boundary layer. The authors brought up earlier measurements from SATURN and southern Finland but missed helicopter/aircraft studies in the remote summertime Arctic (e.g., Kupiszewski et al., ACP 2013; Burkart et al., ACP 2017) that found ultrafine particles near the surface, with much lower concentrations in the free troposphere.

Author reply: We have added discussions regarding measurements in both Amazon basin and summertime Arctic and included the references in the revised version in the Introduction section:

"The rapid downward mixing and transport of small aerosol particles from the free troposphere represents an important source of aerosol particles in the pristine Amazon boundary layer (Wang et al., 2016; Andreae et al., 2017). Helicopter (Kupiszewski et al., 2013) and aircraft (Burkart et al., 2017) measurements of vertical profiles of aerosol particles in the summertime Arctic observed particle formation in the near-surface layer, with much lower concentrations at higher altitudes."

2. Page 4, lines 25-27: What were the explicit size ranges for each CPC? What is the uncertainty associated with the 11-16 nm size window?

[Figure]

Author reply: The minimum cut-off size of the two TSI model 3007 CPCs are 11 and 16 nm respectively. The maximum size is listed as >1.0 $\mu$m. For the counting efficiency curves shown in Figure 1, the accuracy of the measured diameter is determined by the uncertainties in operating voltage and flow rate of the DMA that was used for calibration. We estimate the uncertainty in diameter to be +/- 5%. The absolute uncertainty of particle number concentration, as specified by the manufacturer, is +/- 20%. This results in a propagated uncertainty in the measured ratio of +/- 28%.

We are motivated by these questions to clarify this discussion, and have added the following sentence to the Method section in the revised version. We have also modified Figure 1 and its caption to correctly show these uncertainties.

"The cut-off sizes (defined as the size at which particles are detected with 50% efficiency) are about 11 nm and 16 nm for the CPCs with TEDs operating at 1600 and 1000 mV, respectively. The number concentration of particles in the size range of 11-16 nm (N11-16) was estimated by subtracting measurements from the two CPCs. The time resolution of CPC measurements is one second, and the uncertainty is $\pm$ 20%."

3. Figure 2: The grey and white colors in the legend are reversed.

Author reply: The Figure 2 caption has been corrected in the revised version.

4. Captions for Figures 3 and 5: "Tethered" is misspelled in line 4.

Author reply: The misspellings have been corrected in the revised version.

5. In section 3.1, how representative are the two days for general conditions at the site? How do they compare to the other days simulated in Figure 9?

Author reply: We have limited data to conclude how representative the two days are for general conditions at the SGP site, but we are able to discuss the similarities between the 5 days and, for some observations, compare these days to campaign averages. A comparison can be made of the measured hygroscopicity parameter for 13 nm diameter particles, kappa(13nm), for the two flight days compared to the campaign average

by comparing the open circles in the top plots of Figures 7a and 7b to the data in Figure 6. This is shown, for clarity, in the attached Fig. 1. May 12th, which featured a strong new particle formation event with modest predicted water content, appears to have lower-than-normal values of kappa, typical of organic aerosol, during the daytime. The event on May 13th shows higher kappa that is more representative of the campaign average.

With regard to atmospheric stability, we can only compare the 5 days that were the focus of this study. This can be done by comparing the plots in Figure 2 as well as Figure S1, which together show vertically and temporally resolved atmospheric stability for all 5 days discussed in the manuscript. All days show the general features of a stable boundary layer at night, with a height that is at its minimum at night. May 9th and 13th (Figures S1b and S1d) show another region of stability at ∼500m. It is important to note that the key observation from our stability analysis is that the atmosphere is well-mixed during the daytime when new particle formation is observed. This seems to be a general feature of all days studied. With regard to relative humidity of the atmosphere, that information for the 5 days of this study can be found by comparing Figures 7a and 7b (top plots), with Figures 8, S7 and S8 (middle plots). Clearly, May 9th was the day with the most atmospheric water vapor of all days studied. April 19th, May 12th, and May 13th all showed similar RH characteristics of ∼50% during the daytime.

While we cannot comment as to whether these events are "normal," we agree with the spirit of the reviewer's questions that the reader should be aware of the fact that a more comprehensive analysis of the site would be desireable. We have added the following in the Conclusions:

"For the current study we focus only on two days during which tethered balloon measurements coincided with new particle formation events and three days during which we made a comprehensive analysis of nanoparticle growth rates. While we observed a large range of atmospheric conditions as well as nanoparticle properties in our study, additional investigations are needed to assess how representative these days are for

general conditions at the SGP site."

Reviewer: 2

The paper presents an important topic, the vertical profile of new particle formation and further estimates and speculates on the LWC of these newly formed particles. The paper uses two CPCs of different lower cut-off size for the profile measurements of 11-16 nm particles. Further on the authors take advantage of the ground level SMPS and HTDMA measurements and a model. The paper is clearly written and the presentation of results is well performed explaining the assumptions needed and uncertainties in the analysis. Still, I recommend the authors to consider the following comments.

1. Could you justify the selection of 11-16 nm range for the profile measurements? As the profiles are measured with 11-16 nm particles, the observed particles are the particles that have been growing 1-3 hours after the actual onset of the NPF event itself. Does this affect the analysis and what kind of uncertainty this brings in? Based on the fact above, as the 11-16 nm particles have been first observed at well above ground level, could it be that this is due to a larger growth rate from 1 nm to 11nm at this altitude, and also more intense burst of particles?

Author reply: Regarding the reason for the size range, this is a direct result of the range with which the TSI model 3007 portable condensation particle counters can be adjusted for the minimum detectable particle size. This is adjusted by the voltage applied to the thermoelectric device (TED), and the selection of 1600mV and 1000mV balanced the need for low diameter detection limit with the risk of damaging the instrument by driving the TEDs too hard.

We agree with the reviewer that particles that grow into this detection range of our profile CPCs are likely to have been formed 1-3 hours prior to detection. This could have been mitigated by using particle counters that are light-weight, low-powered, and can detect sub-3 nm diameter particles; however, to our knowledge such particle counters do not currently exist. We also agree with the reviewer that this does add some degree

of uncertainty to the observation that, for the May 12th event, new particle formation was initiated aloft. The reviewer suggests one possible interpretation: that nucleation may have occurred at all levels in the boundary layer, but the growth rate to 11 nm may have been faster at 500 m above ground compared to that directly above ground (or the event was "stronger"). To a certain degree, this scenario would depend extent to which the boundary layer is well-mixed. Our bulk Richardson number analysis of the May 12th event (Figure 2) showed that the boundary layer was moderately to extremely unstable during this time. Atmospheric vertical velocities as measured by the Raman Lidar, which were used to calculate the Richardson number are shown inthe attached Fig. 2 and demonstrate by turbulent the atmosphere was during the daytime at SGP.

Because of the well-mixed boundary layer we feel that the processes that control growth at 500 m are similar to those on the surface. However it is important to note that there is some uncertainty implicit in this statement. Because of this, we will add the following text to the document:

"Implicit in these statements is that the vertical distribution of NPF is adequately represented by the vertical distribution of 11-16 nm diameter particles. Particle formation is estimated to have been initiated 3-4 hours prior to detection by the balloon-borne CPCs, based on the $\sim$3 nm hr-1 diameter growth rate of the May 12th event as determined from the size distributions shown in Figure 3a. Because of this, there is some degree of uncertainty associated with linking these observations to NPF. Deploying CPCs that are light-weight, low-powered, and can detect sub-3 nm diameter particles can eliminate this uncertainty. Such particle counters do not exist currently, but rapid progress is being made in this area."

2. Page 4, rows 3-20: Are the particles measured with the profile CPCs and ground-based SMPSs dry or wet/ambient?

Author reply: The profile CPC particles were measured under ambient conditions. Regarding the ground-based SMPS, we did not use a drier on the sample air. Since the

sheath air of the DMA was recirculated, the dew point in the DMA classifying column was approximately equal to the ambient dew point. Because the trailer was air conditioned, however, the temperature in the DMA column was not always equal to the ambient temperature, so the RH would not have exactly equaled the ambient RH. We chose this sampling arrangement to ensure that the RH of the measured aerosol was close to the ambient RH. In response to this question we have added the following sentence to the text:

"Both SMPSs and CPCs measured particles under ambient conditions."

3. Page 6, row 3-11: Please mention the used HTDMA diameters already here.

Author reply: We have revised the manuscript by adding the following sentence to the Method section,

"The hygroscopic growth factors for particles with diameters of 13, 25, 50, 100 and 200 nm are used for analysis in this study."

4. Page 10, row 24: How much did the hygroscopic growth factor and Kappa-value varied throughout the period and especially on the case days 12-13 May?

Author reply: The temporally resolved hygroscopicity parameter kappa of 13-nm particles on the case days of May 12 and 13 are showed in the upper plots of Figures 7a and 7b and these can be compared to the campaign average kappa shown in Figure 6. Fig. 1, above, also shows these plots together. We choose not include the Fig. 1 in the manuscript since we feel this information is already displayed in Figures 6, 7a, and 7b.

5.Figure 2: The legend and figure caption colors do not match.

Author reply: The Figure 2 caption has been corrected in the revised version.

Please also note the supplement to this comment:
https://www.atmos-chem-phys-discuss.net/acp-2017-586/acp-2017-586-AC1-

supplement.pdf

[Figure]

**Fig. 1.** Response Figure 1: Diurnal plots of kappa for 13 nm diameter particles as measured by the ground-based TDMA, compared to campaign averages. The boxes represent 25-75% ranges of all measurements.

[Figure]

**Fig. 2.** Response Figure 2: Vertical velocity as measured by Raman Lidar during May 12th, 2013.

**Supplement:**

**Vertically resolved concentration and liquid water content of atmospheric nanoparticles at the US DOE Southern Great Plains site**

Haihan Chen1, Anna L. Hodshire2, John Ortega3, James Greenberg3, Peter H. McMurry4, Annmarie G. Carlton1, Jeffrey R. Pierce2, Dave R. Hanson5, James N. Smith\*,1

[revised manuscript text omitted]